# Symptoms and symptom clusters associated with SARS-CoV-2 infection in community-based populations: Results from a statewide epidemiological study

**Brian E. Dixon**[1,2]*, **Kara K. Wools-Kaloustian**[3,4], **William F. Fadel**[5], **Thomas J. Duszynski**[1], **Constantin Yiannoutsos**[5], **Paul K. Halverson**[6,7], **Nir Menachemi**[2,6]

1 Department of Epidemiology, IU Fairbanks School of Public Health, Indianapolis, Indiana, United States of America, 2 Center for Biomedical Informatics, Regenstrief Institute, Indianapolis, Indiana, United States of America, 3 Department of Medicine, IU School of Medicine, Indianapolis, Indiana, United States of America, 4 Center for Global Health, Indiana University, Indianapolis, Indiana, United States of America, 5 Department of Biostatistics, IU Fairbanks School of Public Health, Indianapolis, Indiana, United States of America, 6 Department of Health Policy and Management, IU Fairbanks School of Public Health, Indianapolis, Indiana, United States of America, 7 Department of Family Medicine, IU School of Medicine, Indianapolis, Indiana, United States of America

* bedixon@regenstrief.org

## Abstract

### Background

Prior studies examining symptoms of COVID-19 are primarily descriptive and measured among hospitalized individuals. Understanding symptoms of SARS-CoV-2 infection in pre-clinical, community-based populations may improve clinical screening, particularly during flu season. We sought to identify key symptoms and symptom combinations in a community-based population using robust methods.

### Methods

We pooled community-based cohorts of individuals aged 12 and older screened for SARS-CoV-2 infection in April and June 2020 for a statewide prevalence study. Main outcome was SARS-CoV-2 positivity. We calculated sensitivity, specificity, positive predictive value (PPV), and negative predictive value (NPV) for individual symptoms as well as symptom combinations. We further employed multivariable logistic regression and exploratory factor analysis (EFA) to examine symptoms and combinations associated with SARS-CoV-2 infection.

### Results

Among 8214 individuals screened, 368 individuals (4.5%) were RT-PCR positive for SARS-CoV-2. Although two-thirds of symptoms were highly specific (>90.0%), most symptoms individually possessed a PPV <50.0%. The individual symptoms most greatly associated with SARS-CoV-2 positivity were fever (OR = 5.34, p<0.001), anosmia (OR = 4.08, p<0.001), ageusia (OR = 2.38, p = 0.006), and cough (OR = 2.86, p<0.001). Results from

**Data Availability Statement:** All data used in this study are available from the IUPUI DataWorks repository (https://dx.doi.org/10.7912/D2/21).

**Funding:** This study was funded by the Indiana Department of Health to the Fairbanks School of Public Health (PI - NM) as a component of its efforts to examine seroprevalence of the state population for COVID-19. The funder played a role in data collection, but it did not play a role in study design, analysis, or preparation of the manuscript. Dr. Dixon receives funding from the U.S. National Library of Medicine (T15LM012502) to train public health informatics researchers as well as the U.S. Centers for Disease Control and Prevention (U18DP006500) and the Indiana Department of Health to support disease surveillance research.

**Competing interests:** No authors have competing interests.

EFA identified two primary symptom clusters most associated with SARS-CoV-2 infection: (1) ageusia, anosmia, and fever; and (2) shortness of breath, cough, and chest pain. Moreover, being non-white (13.6% vs. 2.3%, p<0.001), Hispanic (27.9% vs. 2.5%, p<0.001), or living in an Urban area (5.4% vs. 3.8%, p<0.001) was associated with infection.

## Conclusions

Symptoms can help distinguish SARS-CoV-2 infection from other respiratory viruses, especially in community or urgent care settings where rapid testing may be limited. Symptoms should further be structured in clinical documentation to support identification of new cases and mitigation of disease spread by public health. These symptoms, derived from asymptomatic as well as mildly infected individuals, can also inform vaccine and therapeutic clinical trials.

## Introduction

Severe acute respiratory syndrome coronavirus 2 (SARS-CoV-2) causes COVID-19 disease, which has a range of manifestations from asymptomatic infection to severe pneumonia, potentially leading to intensive care utilization or death. Globally there have been more than 102 million cases and over 2.2 million deaths as of January 31, 2021. Further, as of that date, the U.S. and several nations have now experienced three distinct waves of the COVID-19 pandemic. Most recently, cases in the U.S. as well as Europe surged heading into the winter holidays, with many nations re-implementing mitigation techniques to slow the spread of the virus.

Symptoms of COVID-19 are similar to those of influenza and other respiratory diseases. Based primarily on studies of hospitalized individuals, the U.S. Centers for Disease Control and Prevention (CDC) recognized three principal symptoms for COVID-19: fever, cough, and shortness of breath (dyspnea) [1]. This list was expanded as the pandemic progressed to include chills, myalgias, headache, sore throat, and the loss of taste (ageusia) and/or smell (anosmia). An early systematic review that included 1,576 hospitalized COVID-19 patients reported that the most prevalent clinical symptom was fever, followed by cough, fatigue and dyspnea [2]. A later review reported the main clinical symptoms to be fever, cough, fatigue, slight dyspnea, sore throat, headache, conjunctivitis and gastrointestinal issues [3]. In a community-based study involving self-reported symptoms via a mobile app, 10 symptoms—fever, persistent cough, fatigue, shortness of breath, diarrhea, delirium, skipped meals, abdominal pain, chest pain and hoarse voice—were associated with self-reported positive test results in a UK cohort [4]. A Cochrane systematic review [5] identified a total of 27 signs and symptoms for COVID-19.

Importantly, much of the information about common COVID-19 symptoms originate from studies focused on limited populations who presented in a hospital setting [5]. Moreover, testing guidelines in the US, and many other parts of the world, have prioritized symptomatic and high-risk individuals, which further biases available data on symptomology towards those with more severe disease. Most studies further focus exclusively on SARS-CoV-2 positive patients and thus lack an uninfected control group which might also exhibit some level of baseline symptoms. In Menni et al. [4] app users self-reported test results limiting reliability of findings in an uncontrolled study. The Cochrane review suggests the existing evidence on symptoms is "highly variable," and no studies to date assessed combinations of different signs

and symptoms [5]. In summary, there is much that can still be learned about the symptomatology of SARS-CoV-2 infection, especially among community-based individuals who may not require or have not yet presented for clinical care.

We sought to examine the symptoms reported by populations of community-based individuals tested for SARS-CoV-2 in the context of a statewide prevalence study. We examine patterns and groups of symptoms, and we compare individuals who tested positive for active viral infection, using RT-PCR, to those who screened negative. We also examine symptom differences by age, sex, race, ethnicity, and rurality. By better characterizing COVID-19 symptoms, especially among those that may have milder disease, we sought to better identify those symptoms and symptom combinations that are more likely to represent SARS-CoV-2 infection. In addition, understanding the symptomology of community-based infections has the potential to inform the selection of end points for consideration in clinical trials focused on vaccine and therapeutic effectiveness.

## Methods

### Study participants and recruitment

Data derived from two waves of testing in Indiana, conducted in partnership with the state health department, were analyzed for this study. Each wave included two groups: individuals that were randomly selected for invitation-only testing, and individuals from predominantly minority communities encouraged to attend open-testing at sites throughout the state. Wave 1 of testing, described elsewhere [6], occurred at the end of April 2020. Wave 2 occurred in the beginning of June 2020. The random (by invitation-only) sample for each wave was selected from individual state tax records of filers and dependents. Randomly selected individuals (N = 15,495) received a postcard, text message, and phone call inviting them for COVID-19 testing at specific sites set-up across the state by the department of health. In addition, because underrepresented minority groups are more seriously impacted by COVID-19, both waves also conducted targeted nonrandom testing, in conjunction with religious and civic leaders, in select African American and Hispanic communities. All individuals in those communities were encouraged to come to open testing sites on specific days, at locations set-up in those communities. Because the sites were open, passersby could walk-in for testing without an appointment.

In all cases, inclusion criteria were Indiana residency and being 12 years of age or older. Individuals from both random and nonrandom samples were tested regardless of symptoms, prior testing history, or medical history. Testing was available with no out-of-pocket costs to participants.

Recruitment was aided by public announcements by the Governor, the media, and minority community leaders. These messages encouraged individuals who received a postcard, text message, or phone call to show up for testing to aid public health agencies track the spread of the virus. State agencies, including the Indiana Minority Health Coalition, reached out to minority communities to encourage participation. In minority communities, the study team also engaged civic and religious leaders to encourage community members to show up for testing at open sites on specific dates. Both types of testing were components of the state's overall efforts to expand testing capacity and surveillance for COVID-19 in the state for all populations.

### COVID-19 testing and specimen collection

Using Dacron swabs and standard techniques, trained personnel collected nasopharyngeal swabs for RT-PCR analysis. Nasopharyngeal swabs were transferred to the laboratories of Eli

Lilly and Company (Lilly Clinical Diagnostics Lab SARS-CoV-2 test based on the CDC primary set) or Indiana University Health (Luminexon NxTAG CoV Extended Panel or Roche cobas SARS-CoV-2 test) for RT-PCR testing. All laboratories were located in Indianapolis, IN. Test results were reported to participants via a secure website within 1 to 4 days.

## Data collection

Upon arrival to a testing site, each participant was asked to complete a research intake form that included questions about symptoms, health status, and demographics. Using a standardized checklist, participants were asked to indicate presence of symptoms within the past two weeks (14 days). The checklist represented a composite of symptoms reported by either the WHO or the CDC to be associated with COVID-19 [1] and expanded to include additional symptoms reported in the literature [7, 8]. Participants were requested to identify all symptoms present, and those who did not indicate any symptoms were categorized as asymptomatic. Data collected from participants, along with laboratory test results, were captured separately by the state health department and distinguished from other testing sites for the study team.

## Statistical analysis

Because the target populations, identification, and recruitment processes were similar between the two waves, all participants from both waves were pooled for analysis. Descriptive statistics were calculated for all individuals who participated in testing for SARS-CoV-2, stratified by the recruitment method (e.g., random, or nonrandom). Chi-square tests were used to compare various characteristics between the two groups as well as characteristics of individuals testing positive for SARS-CoV-2 versus those testing negative.

Using all available data, we first examined the sensitivity, specificity, positive predictive value (PPV), and negative predictive value (NPV) of each symptom reported by participants. Because specific combinations of symptoms might be useful in screening patients who present to a clinic or hospital, we further explored various symptom combinations. First, we examined the sensitivity, specificity, PPV and NPV of symptom pairs and triplets. All such symptom permutations were examined. To further explore symptom combinations, we employed exploratory factor analysis (EFA) to create symptom groups using positive RT-PCR status as the gold standard for identification of SARS-CoV-2 infection. Factor analysis is a recommended multivariate method for exploring symptoms when symptoms are commonly grouped together in a given etiology [9].

Furthermore, we developed a logistic regression model to examine the relationship between the presence of individual symptoms and positive RT-PCR status, controlling for participant demographics and sample recruitment method. The logistic regression model computes the probability of positive RT-PCR status ranging from 0 to 1. We then generated a Receiver Operating Characteristic (ROC) curve associated with the model by varying the cut-off probability across the range of observed values using the pROC package in R. These are plots of the true-positive (sensitivity) versus the false-positive rate (1 –specificity) of a test, over all possible cut-off points [10]. We also calculated the area under the ROC curve (AUC) as a global measure of the accuracy of the model to predict SARS CoV-2 positivity.

All analyses were performed using R (version 3.6.3). The data used in these analyses are available for reproduction and secondary use through IUPUI DataWorks, a repository for research data for faculty at Indiana University-Purdue University Indianapolis [11]. The dataset DOI is https://dx.doi.org/10.7912/D2/21. This study was determined to be exempt by the

Institutional Review Board (IRB) at Indiana University under the public health surveillance exception.

## Results

We screened a total of 8214 individuals for SARS-CoV-2 including 6326 (77.0%) individuals who were randomly selected and 1888 (23.0%) who were recruited for nonrandom testing from minority communities. A total of 368 individuals (4.5%) were RT-PCR positive for SARS-CoV-2. The characteristics of tested participants are summarized in Table 1. Randomly selected and nonrandom participants were significantly (p<0.001) different on all characteristics. The nonrandom group included more males (45% vs. 41%, p<0.001), non-whites (57.2% vs. 8.5%, p<0.001), Hispanics (27% vs. 2.3%, p<0.001), and urban (80.6% vs. 65%, p<0.001) residents. Males and females tested positive in similar proportions (4.6% vs. 4.4%, p = 0.707). Non-whites (13.6% vs. 2.3%, p<0.001), Hispanics (27.9% vs. 2.5%, p<0.001), and individuals living in Urban areas (5.4% vs. 3.8%, p<0.001) had higher rates of positivity.

Symptoms experienced by individuals who tested positive or negative and the sensitivity, specificity, PPV, and NPV for each individual symptom with respect to predicting RT-PCR positivity are summarized in Table 2. Also presented are combinations of symptoms with a PPV ≥60.0%. For a patient to be included in a given combination (e.g., Fever & Loss of Taste), he or she had to report all symptoms part of the combination. Although two-thirds of the symptoms were highly specific for COVID-19 (>90.0%), most symptoms individually possessed a PPV of <50.0%. The three symptoms with the largest individual PPVs were anosmia (52.5%), ageusia (51.0%), and fever (47.6%). When fever was paired with anosmia or ageusia, with or without the presence of a third symptom, the PPV increased to >70%. However, sensitivity for all individual symptoms and symptom combinations was low (<50.0%).

**Table 1. Characteristics of Indiana residents who were tested for SARS-CoV-2 infection in late April and early June 2020, including individuals randomly selected for testing along with nonrandom, targeted populations designed to enhance diversity of the populations tested.**

| Characteristics | Total Participants No. (Col %) | Randomly Selected Individuals No. (Col %) | Individuals from Open Community Testing Locations No. (Col %) | P Value[a] for Group | Individuals Testing Positive via RT-PCR No. (Row %) | P Value[b] for Positivity |
|---|---|---|---|---|---|---|
| Overall Population Totals | 8214 | 6326 | 1888 | | 368 | |
| Female | 4565 (55.6%) | 3451 (54.6%) | 1114 (59.0%) | <0.001 | 201 (4.4%) | 0.71 |
| Male | 3647 (44.4%) | 2873 (45.4%) | 774 (41.0%) | | 167 (4.6%) | |
| White | 6599 (80.3%) | 5791 (91.5%) | 808 (42.8%) | <0.001 | 149 (2.3%) | <0.001 |
| Non-white | 1615 (19.7%) | 535 (8.5%) | 1080 (57.2%) | | 219 (13.6%) | |
| Hispanic | 653 (7.9%) | 144 (2.3%) | 509 (27.0%) | <0.001 | 182 (27.9%) | <0.001 |
| Non-Hispanic | 7561 (92.1%) | 6182 (97.7%) | 1379 (73.0%) | | 186 (2.5%) | |
| Urban[c] | 5631 (68.6%) | 4109 (65.0%) | 1522 (80.6%) | <0.001 | 303 (5.4%) | <0.001 |
| Rural/Mixed | 1598 (19.5%) | 1500 (23.7%) | 98 (5.2%) | | 27 (1.7%) | |
| Rural | 980 (11.9%) | 717 (11.3%) | 263 (13.9%) | | 37 (3.8%) | |
| Age: <40 | 2379 (29.0%) | 1747 (27.6%) | 632 (33.5%) | <0.001 | 167 (7.0%) | <0.001 |
| Age: 40–59 | 3036 (37.0%) | 2308 (36.5%) | 728 (38.6%) | | 155 (5.1%) | |
| Age: 60+ | 2799 (34.1%) | 2271 (35.9%) | 528 (28.0%) | | 46 (1.6%) | |

[a]Comparison of Randomly Selected individuals to nonrandom community testing.

[b]Comparison of Individuals Testing Positive to those Testing Negative.

[c]Based upon Purdue Rural Indiana Classification System.

**Table 2. Self-reported symptoms by participants undergoing SARS-CoV-2 testing in a statewide prevalence study.** All individual symptoms and lack of symptoms (asymptomatic) included as well as combinations of symptoms with a positive predictive value >60%.

| Symptoms reported in the past 14 days | Number of individuals testing positive via RT-PCR N (%) | Number of individuals testing negative via RT-PCR N (%) | Sensitivity | Specificity | Positive Predictive Value | Negative Predictive Value |
|---|---|---|---|---|---|---|
| Overall Population Totals | 368 (4.6) | 7650 (95.4) | | | | |
| Asymptomatic (no reported symptoms) | 91 (24.7) | 4681 (61.2) | 24.7% | 38.8% | 1.9% | 91.5% |
| **Individual Symptoms** | | | | | | |
| Loss of Smell (anosmia) | 94 (25.5) | 85 (1.1) | 25.5% | 98.9% | 52.5% | 96.5% |
| Loss of Taste (ageusia) | 105 (28.5) | 101 (1.3) | 28.5% | 98.7% | 51.0% | 96.6% |
| Fever | 129 (35.1) | 142 (1.9) | 35.1% | 98.1% | 47.6% | 96.9% |
| Chills | 81 (22) | 197 (2.6) | 22.0% | 97.4% | 29.1% | 96.3% |
| Chest Pain | 65 (17.7) | 255 (3.4) | 17.7% | 96.6% | 20.3% | 96.0% |
| Vomiting | 12 (3.3) | 51 (0.7) | 3.3% | 99.3% | 19.0% | 95.5% |
| Muscle Ache (myalgia) | 117 (31.8) | 564 (7.4) | 31.8% | 92.6% | 17.2% | 96.6% |
| Cough | 175 (47.6) | 969 (12.7) | 47.6% | 87.3% | 15.3% | 97.2% |
| Shortness of Breath | 64 (17.4) | 438 (5.8) | 17.4% | 94.2% | 12.7% | 95.9% |
| Sore Throat | 90 (24.5) | 618 (8.1) | 24.5% | 91.9% | 12.7% | 96.2% |
| Diarrhea | 59 (16) | 479 (6.3) | 16.0% | 93.7% | 11.0% | 95.8% |
| Fatigue | 127 (34.5) | 1063 (14) | 34.5% | 86.0% | 10.7% | 96.4% |
| Headache | 144 (39.1) | 1376 (18.1) | 39.1% | 81.9% | 9.5% | 96.5% |
| Runny Nose | 90 (24.5) | 1116 (14.7) | 24.5% | 85.3% | 7.5% | 95.9% |
| **Symptom Combinations** | | | | | | |
| Fever & Loss of Taste (ageusia) | 63 (17.1) | 24 (0.3) | 17.1% | 99.7% | 72.4% | 96.1% |
| Fever & Loss of Smell (anosmia) | 53 (14.4) | 22 (0.3) | 14.4% | 99.7% | 70.7% | 96.0% |
| Loss of Taste (anosmia) & Vomiting | 8 (2.2) | 4 (0.1) | 2.2% | 99.9% | 66.7% | 95.5% |
| Cough & Fever & Loss of Taste (ageusia) | 49 (13.3) | 13 (0.2) | 13.3% | 99.8% | 79.0% | 96.0% |
| Cough & Fever & Loss of Smell (anosmia) | 41 (11.1) | 13 (0.2) | 11.1% | 99.8% | 75.9% | 95.9% |
| Fever & Loss of Smell (anosmia) & Muscle Ache (myalgia) | 37 (10.1) | 13 (0.2) | 10.1% | 99.8% | 74.0% | 95.8% |
| Fever & Loss of Taste (ageusia) & Muscle Ache (myalgia) | 46 (12.5) | 18 (0.2) | 12.5% | 99.8% | 71.9% | 95.9% |
| Fever & Loss of Smell (anosmia) & Loss of Taste (ageusia) | 45 (12.2) | 18 (0.2) | 12.2% | 99.8% | 71.4% | 95.9% |
| Fever & Headache & Loss of Smell (anosmia) | 42 (11.4) | 17 (0.2) | 11.4% | 99.8% | 71.2% | 95.9% |
| Fatigue & Fever & Loss of Smell (anosmia) | 36 (9.8) | 15 (0.2) | 9.8% | 99.8% | 70.6% | 95.8% |
| Chills & Fever & Loss of Smell (anosmia) | 26 (7.1) | 11 (0.1) | 7.1% | 99.9% | 70.3% | 95.7% |
| Diarrhea & Fever & Loss of Taste (ageusia) | 23 (6.2) | 10 (0.1) | 6.2% | 99.9% | 69.7% | 95.7% |
| Chills & Fever & Loss of Taste (ageusia) | 36 (9.8) | 16 (0.2) | 9.8% | 99.8% | 69.2% | 95.8% |
| Fever & Headache & Loss of Taste (ageusia) | 47 (12.8) | 21 (0.3) | 12.8% | 99.7% | 69.1% | 95.9% |
| Fatigue & Fever & Loss of Taste (ageusia) | 42 (11.4) | 19 (0.2) | 11.4% | 99.8% | 68.9% | 95.9% |

We further explored symptom combinations in which the patient reported one or more symptoms included in the combination (e.g., Fever or Loss of Taste, Chills or Fever or Diarrhea). These results are included as S1 Table. Unlike Table 2, patients included in S1 Table were included if they reported at least one of the symptoms listed, even if they were absent the other(s) indicated in the combination. Changing the operator from an AND to an OR increased sensitivity, but specificity and PPV decreased. None of the PPVs exceeded 50%. For example, whereas the combination of fever and ageusia possessed a sensitivity of 17.1% and a PPV of 72.4%, a scenario where the patient reported fever or ageusia resulted in a sensitivity of 46.5% but a PPV of 43.8%.

## Symptom groups

The principal symptom groups identified through EFA are summarized in Fig 1. Five principal symptom groups emerged, all of which have face validity. The first symptom group (Factor 1) consists of ageusia (r = 0.92), anosmia (r = 0.90), fever (r = 0.63). The second symptom group (Factor 2) consists of shortness of breath (r = 0.83), cough (r = 0.49), and chest pain (r = 0.64). The third symptom group (Factor 3) consists of fatigue (r = 0.73) and myalgias (r = 0.71). The fourth symptom group (Factor 4) consists of vomiting (r = 0.90) and diarrhea (r = 0.55). The final symptom group (Factor 5) consists of runny nose (r = 0.80) and sore throat (r = 0.44). The cumulative variance explained by all symptom groups was 70%, with the first two groups (Factors 1 and 2) explaining 49% of the variance.

## Predicting COVID-19 positivity based on symptoms and demographics

The results of the logistic regression model that examined how individual symptoms were associated with RT-PCR positivity are presented in Table 3. When controlling for demographics as well as testing group (e.g., random vs. nonrandom sample) and other symptoms, individual symptoms most strongly associated with high odds of RT-PCR SARS-CoV-2 positivity were fever (OR = 5.34, p<0.001), anosmia (OR = 4.08, p<0.001), ageusia (OR = 2.38, p = 0.006), and cough (OR = 2.86, p<0.001). The ROC curve generated by a test based on this cluster of symptoms is presented in Fig 2. The AUC for the diagnostic index generated by the model which included this symptom cluster was 0.909. Individuals from the nonrandom sample were more likely than randomly selected participants to test positive (OR = 9.34, p<0.001). Hispanic participants were more likely (OR = 2.44, p<0.001) to test positive than non-Hispanic participants. Older participants, age ≥60 years, were less likely (OR = 0.43, p<0.001) to test positive compared to participants under 40 years of age.

## Discussion

In this study of populations screened for COVID-19, we observed a wide constellation of symptoms present among those testing positive for active infection. Many symptoms commonly reported by infected participants (e.g., cough, fatigue, headache, myalgias) were similar to those identified by the CDC [1] and those reported in prior studies [3, 7, 8, 12–14]. However, unlike prior studies, we also captured symptoms from those testing negative, enabling the examination of the sensitivity, specificity, PPV and NPV of individual symptoms and symptom combinations.

Our findings provide robust evidence to the growing body of studies that identify anosmia and ageusia as important symptoms of SARS-CoV-2 infection [12, 15, 16]. These symptoms individually are highly specific for COVID-19 infection and, in combination with fever, are highly predictive of positive status. Therefore, presence of these symptoms should trigger use of personal protective equipment (PPE), if universal PPE is not already being utilized, and

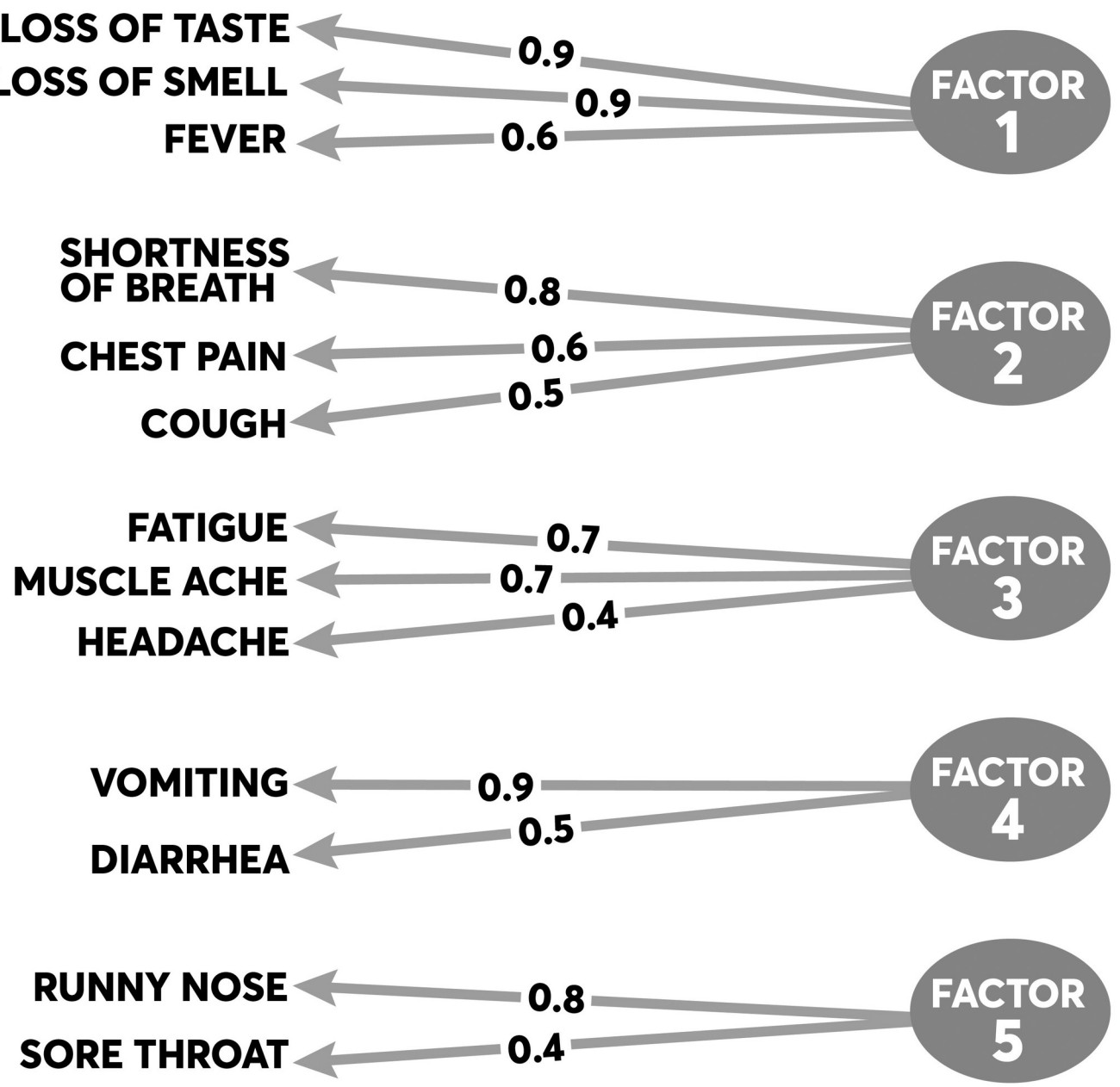

**Fig 1. Results from the exploratory factor analysis of symptom clusters.** The figure summarizes the principal symptoms loading on each factor. The proportions represent individual loadings for each symptom onto its factor based on the factor's correlation matrix.

prompt testing for COVID-19. Furthermore, particularly during influenza season, it is important that health care providers specifically ask questions about loss of taste and smell in addition to traditional questions asked for assessing acute respiratory illness. Moreover, documenting these symptoms can aid public health officials in distinguishing COVID-19 from influenza within syndromic surveillance systems [17]. Currently syndromes recommended by CDC and others overlap with influenza-like illness, and few electronic health records (EHR) systems capture information about loss of taste or smell.

**Table 3. Logistic regression model to predict SARS-CoV-2 RT-PCR positivity using participant reported symptoms and demographics.**

| Symptom | AOR[1] | 95% CI[1] | p-value |
|---|---|---|---|
| Fever | 5.34 | 3.51, 8.12 | <0.001 |
| Loss of Smell (Anosmia) | 4.08 | 2.14, 7.74 | <0.001 |
| Loss of Taste (Ageusia) | 2.38 | 1.28, 4.45 | 0.01 |
| Cough | 2.86 | 2.06, 3.97 | <0.001 |
| Shortness of Breath | 0.61 | 0.36, 1.00 | 0.05 |
| Chest Pain | 1.00 | 0.60, 1.64 | >0.999 |
| Muscle Ache (Myalgia) | 1.07 | 0.70, 1.61 | 0.75 |
| Fatigue | 1.25 | 0.84, 1.82 | 0.27 |
| Headache | 0.75 | 0.52, 1.07 | 0.12 |
| Diarrhea | 1.00 | 0.62, 1.58 | 0.99 |
| Vomiting | 1.35 | 0.47, 3.60 | 0.56 |
| Sore Throat | 0.80 | 0.53, 1.18 | 0.27 |
| Runny Nose | 1.14 | 0.79, 1.63 | 0.46 |
| **Demographics** | | | |
| Age: <40 | — | — | |
| Age: 40–59 | 0.88 | 0.66, 1.18 | 0.41 |
| Age: 60+ | 0.43 | 0.28, 0.63 | <0.001 |
| **Male** | 1.19 | 0.91, 1.56 | 0.21 |
| **Race** | | | |
| White | — | — | |
| Black or African American | 0.80 | 0.49, 1.26 | 0.34 |
| Other | 1.68 | 1.17, 2.41 | 0.01 |
| **Hispanic** | 2.44 | 1.68, 3.54 | <0.001 |
| **Geography** | | | |
| Urban | — | — | |
| Rural | 1.53 | 0.97, 2.37 | 0.06 |
| Rural/Mixed | 1.30 | 0.79, 2.06 | 0.29 |
| **Participation** | | | |
| Random Sample | — | — | |
| Nonrandom Sample | 9.35 | 6.59, 13.4 | <0.001 |

[1] AOR = Adjusted Odds Ratio, CI = Confidence Interval.

We further observed a high rate of asymptomatic or pre-symptomatic individuals with 44.2% of the randomly selected participants and 20.2% of the nonrandom participants reporting a lack of related symptoms within two weeks of their positive RT-PCR test [6]. A high rate of asymptomatic infection likely influenced the overall low PPV and sensitivity for individual symptoms and symptom combinations. The high proportion of asymptomatic cases complicates identification, control, and containment of new SARS-CoV-2 infections by public health authorities. Virtual health care screening as well as chatbots and mobile applications [18, 19] are unlikely to refer asymptomatic individuals for testing. Without sufficient capacity for rapid, inexpensive testing, screening efforts may be hampered and accurate measurement of incidence and prevalence will be challenging.

As observed in other studies that analyzed symptomatic cases [20, 21], we found higher infection rates among Hispanic and non-white populations. These results were largely influenced by individuals involved in the nonrandom samples, who presented to community-based

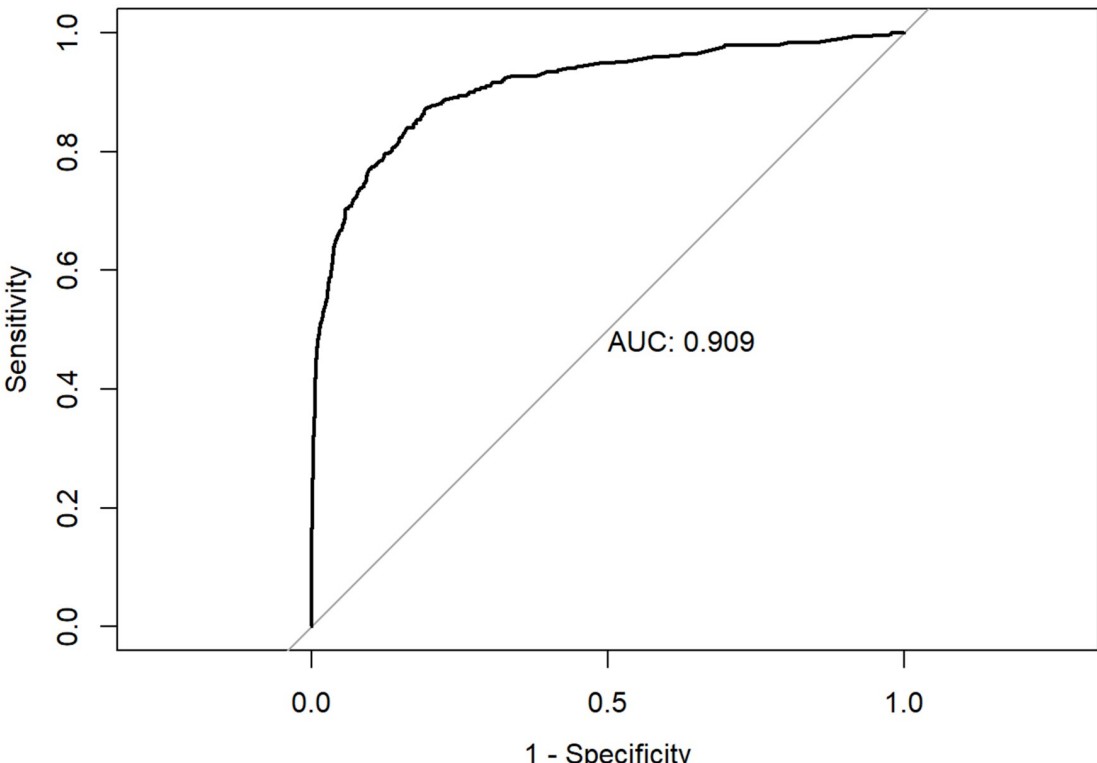

**Fig 2. Area under the curve (AUC) for logistic model fit to predict RT-PCR status given symptoms or patient demographics.**

testing sites. Higher burden of disease among people of color have been widely reported and may stem from systemic inequities in economic, physical, and emotional health (e.g., social determinants) [22]. Targeted efforts to screen and test minority and other underserved populations, in coordination with public health and community-based organizations, are needed to adequately identify and reduce COVID-19 disparities.

## Limitations

We acknowledge several limitations. First, the two testing waves occurred as influenza season waned, which may decrease the presence of influenza-like-illness symptoms among those testing negative for COVID-19. Second, the association between age and RT-PCR positivity, e.g., older participants (60+ years) were less likely to test positive, may be driven by non-response to the sampling by these groups, as elderly and frail individuals may not have been as likely to participate in our studies as they were either tested in a nursing home (a population excluded from our sample) or were unlikely to present to a testing site. It is likely that the differences observed in positivity among non-white and Hispanic populations are due to their self-selecting into testing at the nonrandom sites. They might have been more symptomatic and therefore motivated to seek testing as opposed to the randomized individuals. Finally, although the population was representative of Indiana, the cohort might not represent other states or the nation.

Despite these limitations, the study possessed significant strengths. The cohort was large and diverse, and it was drawn from a statewide population. Second, the symptoms examined were broad and included the most complete list based on available evidence. Finally, the robust methods yielded a strong model with face validity.

## Conclusion

This study finds that key symptoms for identifying active SARS-CoV-2 infection are anosmia and ageusia, especially in association with fever. Cough, especially when present along with shortness of breath and chest pain, is also an important symptom when diagnosing COVID-19. When laboratory testing is not accessible, these symptoms may help guide distinguishing COVID-19 from influenza-like illness. These symptoms should be used for screening and incorporated into clinical documentation to support public health identification of new cases and mitigation of disease spread.

## Supporting information

**S1 Table. Self-reported symptom combinations by participants undergoing SARS-CoV-2 testing.** Patients were considered to have a given combination if they reported one or more of the listed symptoms. Only combinations with a positive predictive value >20% were included. (DOCX)

## Acknowledgments

We thank our collaborators at the Indiana Department of Health, the Governor's Office, the Indiana Management Performance Hub, as well as the Family Social Services Administration for their support of the statewide testing study and data management that enabled this research. We further thank the librarians at the IUPUI University Library for their assistance with preparing and submitting the dataset for archival and retrieval via the Data-Works platform.

## Author Contributions

**Conceptualization:** Brian E. Dixon, Constantin Yiannoutsos, Paul K. Halverson, Nir Menachemi.

**Data curation:** William F. Fadel, Constantin Yiannoutsos.

**Formal analysis:** Brian E. Dixon, William F. Fadel, Constantin Yiannoutsos.

**Funding acquisition:** Paul K. Halverson, Nir Menachemi.

**Investigation:** Brian E. Dixon, Kara K. Wools-Kaloustian, William F. Fadel, Thomas J. Duszynski, Constantin Yiannoutsos, Paul K. Halverson, Nir Menachemi.

**Methodology:** Brian E. Dixon, Kara K. Wools-Kaloustian, William F. Fadel, Thomas J. Duszynski, Constantin Yiannoutsos, Nir Menachemi.

**Project administration:** Paul K. Halverson, Nir Menachemi.

**Resources:** Paul K. Halverson, Nir Menachemi.

**Supervision:** Brian E. Dixon, Kara K. Wools-Kaloustian, Paul K. Halverson.

**Validation:** Kara K. Wools-Kaloustian, William F. Fadel, Constantin Yiannoutsos.

**Visualization:** Brian E. Dixon.

**Writing – original draft:** Brian E. Dixon.

**Writing – review & editing:** Brian E. Dixon, Kara K. Wools-Kaloustian, William F. Fadel, Thomas J. Duszynski, Constantin Yiannoutsos, Paul K. Halverson, Nir Menachemi.

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
