## [Decision Letter · Decision Letter 0]

11 Jan 2021

PONE-D-20-33594

Symptoms and symptom clusters associated with SARS-CoV-2 infection in community-based populations: Results from a statewide epidemiological study

PLOS ONE

Dear Dr. Dixon,

Thank you for submitting your manuscript to PLOS ONE. After careful consideration, we feel that it has merit but does not fully meet PLOS ONE’s publication criteria as it currently stands. Therefore, we invite you to submit a revised version of the manuscript that addresses the points raised during the review process.

We look forward to receiving your revised manuscript.

Kind regards,

Nickolas D. Zaller

Academic Editor

PLOS ONE

Journal Requirements:

Reviewers' comments:

Reviewer's Responses to Questions

**Comments to the Author**

1. Is the manuscript technically sound, and do the data support the conclusions?

Reviewer #1: Partly

Reviewer #2: Yes

2. Has the statistical analysis been performed appropriately and rigorously? 

Reviewer #1: I Don't Know

Reviewer #2: Yes

3. Have the authors made all data underlying the findings in their manuscript fully available?

Reviewer #1: Yes

Reviewer #2: No

4. Is the manuscript presented in an intelligible fashion and written in standard English?

Reviewer #1: Yes

Reviewer #2: Yes

5. Review Comments to the Author

Reviewer #1: PLOS One Manuscript Review

This study assessed symptoms and symptom clusters associated with SARS-CoV-2 in a statewide epidemiological study.

Introduction:

-Update global cases and deaths as they have surpassed 54 million & 1.3 million as of Nov. 16

-End of first paragraph will need to be reworked since fall/winter is here and COVID is spiking again.

-Paragraph 2, calling COVID-19 an “influenza-like illness” could be confusing given the misconception among some that it is the same virus as flu or has the same severity of flu.

-Clarify that dyspnea is shortness of breath when shortness of breath first mentioned like is done with loss of taste (ageusia) and smell (anosmia).

-Anything updated about cardiovascular effects or are those not symptoms that are identified outside of a healthcare setting?

Methods:

-Were the participants mentioned who were not randomly selected and sought testing in open-testing sites the same participants who were recruited by civic leaders in African American and Hispanic communities? First paragraph of methods mentions random selection, those seeking open-testing sites and nonrandom testing of underrepresented groups. Clarify if 2 are the same.

-For groups that were randomly selected, were they asked to be tested? How was that testing accessed and tracked?

Results:

-With the findings being so different between the random and nonrandom sample, it is difficult to interpret

Reviewer #2: This is a well-done timely study of interest any practicing clinicians health officials.

Methods-

Study Participants and Recruitment- How many people were randomly selected and how many in the group that presented on their own (is this the same group that was recruited or is this another distinction)? If they are the same then the terminology should be consistent- random selection vs recruitment (instead of switching to "open community testing" in Table 1). If the groups are not same, Were the characteristics of these groups compared statistically?

.How was the randomization undertaken?

Testing- How were the samples distributed to the two labs? Any difference in viral load detection ability in the two labs? Were random samples sent to one lab and recruited to the other?

Results

Table 1-Was the percent of positive tests different in the random group vs the recruited group?

For Table 2- calculating negative predictive value of symptom combinations - Does this mean all three were absent or one of the symptoms was absent, to be considered negative?

The authors should also present the data with sensitivity for symptoms groupers with "or" function to demonstrate that if certain group of symptoms are all absent the sensitivity is higher. Thus can show that a screening list of certain symptoms will work well to screen individuals at workplaces and community settings (similar to the CDC list). Although not required, it will definitely make the study even more interesting.

For Table 3- Those presenting on their own are more likely to have had symptoms with higher positivity. These were also predominantly ethnic minorities. The higher odds ratio in Hispanic patients may be strongly influenced by this fact and should be interpreted with caution. The authors have mentioned this in the limitations.

Discussion

Second para- "These symptoms are highly predictive for positive status" should read "These symptoms individually are highly specific for COVID and in combination with fever are highly predictive for positive status".

6. PLOS authors have the option to publish the peer review history of their article (what does this mean?). If published, this will include your full peer review and any attached files.

Reviewer #1: No

Reviewer #2: **Yes: **Anupam Sule

---

## [Author Response · Author response to Decision Letter 0]

5 Feb 2021

Responses to the reviewers are provided in a separate file uploaded with the manuscript and other files for re-submission. We thank the editors and reviewers for their constructive feedback.

---

## [Decision Letter · Decision Letter 1]

1 Mar 2021

Symptoms and symptom clusters associated with SARS-CoV-2 infection in community-based populations: Results from a statewide epidemiological study

PONE-D-20-33594R1

Dear Dr. Dixon,

We’re pleased to inform you that your manuscript has been judged scientifically suitable for publication and will be formally accepted for publication once it meets all outstanding technical requirements.

Kind regards,

Nickolas D. Zaller

Academic Editor

PLOS ONE

Additional Editor Comments (optional):

Reviewers' comments:

Reviewer's Responses to Questions

**Comments to the Author**

1. If the authors have adequately addressed your comments raised in a previous round of review and you feel that this manuscript is now acceptable for publication, you may indicate that here to bypass the “Comments to the Author” section, enter your conflict of interest statement in the “Confidential to Editor” section, and submit your "Accept" recommendation.

Reviewer #1: All comments have been addressed

Reviewer #2: All comments have been addressed

2. Is the manuscript technically sound, and do the data support the conclusions?

Reviewer #1: Yes

Reviewer #2: Yes

3. Has the statistical analysis been performed appropriately and rigorously? 

Reviewer #1: Yes

Reviewer #2: Yes

4. Have the authors made all data underlying the findings in their manuscript fully available?

Reviewer #1: Yes

Reviewer #2: Yes

5. Is the manuscript presented in an intelligible fashion and written in standard English?

Reviewer #1: Yes

Reviewer #2: Yes

6. Review Comments to the Author

Reviewer #1: (No Response)

Reviewer #2: Dear Authors

Thank you for addressing our concerns. This study will help clinicians managing COVID cases. Good luck

7. PLOS authors have the option to publish the peer review history of their article (what does this mean?). If published, this will include your full peer review and any attached files.

Reviewer #1: No

Reviewer #2: No

---

## [Editor Report · Acceptance letter]

16 Mar 2021

PONE-D-20-33594R1 

Symptoms and symptom clusters associated with SARS-CoV-2 infection in community-based populations: Results from a statewide epidemiological study 

Dear Dr. Dixon:

I'm pleased to inform you that your manuscript has been deemed suitable for publication in PLOS ONE. Congratulations! Your manuscript is now with our production department. 

Kind regards, 

on behalf of

Dr. Nickolas D. Zaller 

Academic Editor

PLOS ONE